# Reinforcement of Frictional Vibration Noise Reduction Properties of a Polymer Material by PTFE Particles

**DOI:** 10.3390/ma15041365

**Published:** 2022-02-12

**Authors:** Naner Li, Conglin Dong, Yuhang Wu

**Affiliations:** 1Department of Engineering Mathematics, University of Bristol, Bristol BS8 1TR, UK; md20342@bristol.ac.uk; 2Reliability Engineering Institute, National Engineering Research Center for Water Transportation Safety, Wuhan University of Technology, Wuhan 430063, China; yhwu@whut.edu.cn

**Keywords:** PTFE, wear, self-lubricating, frictional vibration and noise

## Abstract

The insufficient lubrication of the contact interface between moving parts can easily induce severe friction vibration and noise under extreme working conditions, which will threaten the service life and reliability of polymer moving components, including water-lubricated stern tube polymer bearings. Thermoplastic polyurethane (TPU) modified by polytetrafluoroethylene (PTFE) particles was developed. The effects of PTFE on the wear and vibration behaviors of modified TPU sliding against the ZCuSn_10_Zn_2_ ring-plates were investigated. The coefficients of friction (COFs), wear mass losses, wear morphologies, frictional vibration, and noise were analyzed synthetically. The results showed that a suitable mass content of PTFE reduced the COFs and wear mass losses of the TPU composites by more than 50% and 40%, respectively, while presenting an excellent friction reduction. The lower COFs of modified TPU showed a small fluctuation amplitude and eliminated vibration waveforms at high vibrational frequencies, which was useful for reducing frictional vibration and noise. The knowledge gained in this study is useful for a better understanding of the wear behaviors of polymer composites, as well as for the design a new polymer material with good self-lubricating and frictional vibration and noise reduction properties.

## 1. Introduction

The concepts of “environmentally friendly” and “green ship” have been deeply rooted in people’s hearts [1]. The International Maritime Organization and many classification societies have issued various specifications and standards to promote the development of green ship technologies to ensure that natural water resources avoid as much as possible pollution due to the leakage of lubricating oil [2]. The propulsion system forms the ship’s core, and the stern tube bearings are the main components of the ship’s propulsion system [3,4]. Water-lubricated bearings are typical environmentally friendly mechanical components without oil pollution, which have broad application prospects in the marine field. As a green and pollution-free material, thermoplastic polyurethane has the characteristics of being highly resistant to fatigue and wear, along with low water swelling and good chemical stability [5]. Moreover, it is easy to be processed into key parts using the heat injection molding method due to its low melting temperature, which is between 190 °C and 200 °C [6]. Therefore, it is widely used in the manufacture of water-lubricated polymer bearings in the marine field [7,8]. With the development of the economy, ships are becoming larger, leading to the diameters of the ship’s propulsion shafts becoming bigger and bigger, especially for big ships traveling over seawater. Thus, there is increased demand for reliable and safe ship shaft support systems, including water-lubricated polymer stern tube bearings. The tribological properties of water-lubricated polymer stern tube bearings is one of the key factors affecting the safety and reliability of a ship’s propulsion system [9]. Actually, when large propulsion shafts are subjected to heavy loads, low speeds, turning, and starting or stopping states, hybrid lubrication, boundary lubrication, or dry friction occurs between the wear surfaces of polymer bearings and the propulsion shafts, which results in severe friction processes, eventually causing large frictional forces [10,11,12,13]. Thus, the materials at the contact surfaces of polymer bearings deform easily along the friction direction, leading to an unstable contact state, which causes violent fluctuations in the friction forces [14,15] and eventually generates frictional vibrations and noise [16]. Thus, the safety and reliability of the ship’s propulsion system and the comfort of the ship are compromised. Decreasing the friction forces and their fluctuation amplitudes, as well as stabilizing friction systems, are the key approaches to solving the above problems.

Modifying a polymer material by improving its self-lubricating properties is an effective method to reduce the friction forces and their fluctuation amplitudes [17]. Many scholars have used nanoparticles to enhance the self-lubricating properties of polymers [18,19,20,21,22]. However, the inconsistency between the inorganic fillers and the polymers weakens the other properties of polymers, such as the tearing property, tensile strength, and elongation at break [23]. As is well known, polytetrafluoroethylene (PTFE) is an excellent additive with good self-lubricating properties [24]. The melting temperature of PTFE is higher than 300 °C, allowing it to maintain good mechanical properties at the high temperature of 260 °C [25,26,27]. It is usually formed via a sintering process [28]. Typically, it cannot be directly introduced into ship water-lubricated polymer bearings because of its poor wear resistance. However, these characteristics enable the use of PTFE as an additive in the thermoplastic molding process of low-melting-point composite materials, enabling good self-lubricating properties during processing. This study aimed to modify TPU using PTFE particles, with the goal of improving the self-lubricating properties of TPU under low speeds and heavy loads. The results of this study can be used for the future design of polymer matrix composites with good self-lubricating properties and ideal frictional vibration and noise reduction performance in marine applications.

## 2. Methods and Experiments

### 2.1. Experimental Materials

Commercial thermoplastic polyurethane (TPU) particles (1193A15, D-BASF Company, Ludwigshafen, Germany) with molecular weights ranging from 5 × 10^4^ to 20 × 10^4^ were chosen as the matrix material according to their good thermoplastic properties, as shown in Figure 1a; their diameters were about 2.0 mm. Commercial PTFE particles supplied by the Aladdin Co., Ltd. (Shanghai, China) were chosen as the filler to modify the TPU, as shown in Figure 1c. The micro-morphologies of the PTFE particles are displayed in Figure 1d, revealing a short rod shape with a diameter of approximately 16 ± 5 μm and a length ranging from 30 μm to 150 μm, with aspect ratios ranging from 2 to 15. Their molecular weights ranged from 100 × 10^4^ to 300 × 10^4^. Commercial carbon black (N772, Aladdin Co., Ltd., Shanghai, China) was chosen as the filler to modify the TPU, as shown in Figure 1b, and its grain diameter ranged from 10 nm to 50 nm. It played a role in enhancing the mechanical properties of TPU, and its mass content was set at a constant value of 10%. The distribution maps of fluorine and carbon are visualized using a scanning electron microscope under 1000× magnification in Figure 1d. The physical properties of TPU and PTFE are shown in Table 1 [29,30,31]. Different mass contents of PTFE particles were used to improve the self-lubricating properties of TPU, set to 0% (no PTFE), 3%, 6%, 9%, and 12%. The formulas of the modified TPU composites are shown in Table 2. The mixtures of TPU particles, PTFE particles, and carbon black powder were combined using a twin-screw mixer machine with an L/D ratio of 20 (SHJ-72, Nanjing Jieya Extrusion Equipment Co., Ltd., Nanjing, China). The mixing temperature was set to 210 °C, and the mixing time was set to 15 min. Then, the mixed particles were made into ring-plate-shaped TPU specimens using a custom-made grinding tool by a DKM injection machine (DQ-180T, Qunde Machine Co., Ltd., Suzhou, China) at a melting temperature of approximately 230 °C and an injecting pressure of 130 MPa for 60 s. The internal diameter, outer diameter, thickness, and cross-sectional area of the ring-plates were 39 mm, 44 mm, 8 mm, and 32.58 × 10^−5^ m^2^, respectively, as shown in Figure 1e. The thermal gravity curve of modified TPU with 12% PTFE and 10% carbon black was examined using a simultaneous thermal analyzer (STA449F3, Netzsch, Bavaria, Germany) to check the real mass contents of the modified composites. The heating rate was 5 °C/min, yielding the results shown in Figure 2. From 260 °C to 500 °C, the mass of the modified composite was reduced by 76.8%, similar to the mass ratio of TPU in the modified composite shown in Table 2. A further reduction by 11.4% brought it close to the mass ratio of PTFE in the modified composite. A further increase in temperature resulted in an additional 7.7% reduction, leaving a residual mass ratio of 4.1%. The modified TPU specimens were polished before the test, and their surface roughness was determined to be approximately 500 ± 100 nm by confocal laser scanning microscopy (CLSM) (VX-X1000, KEYENCE, Ōsaka, Japan). Figure 1f is the microscopic optical photograph of the TPU surface modified by 6% PTFE particles (OLYMPUS BX51, Tokyo, Japan) under 500× magnification, showing the clear distribution of PTFE particles on the surface. Due to 230 °C being much lower than the melting temperature of PTFE, as shown in Table 1, the PTFE particles could keep their original form in the modified TPU, thereby presenting their excellent lubrication performance. ZCuSn_10_Zn_2_ has excellent corrosion resistance and chemical stability in seawater, and it is usually made into a sleeve which wraps the outer surface of the propulsion shaft to avoid seawater corrosion. In fact, water-lubricated polymer stern tube bearings usually slide against ZCuSn_10_Zn_2_ sleeves during the operation of propulsion systems. Thus, ZCuSn_10_Zn_2_ ring-plates were developed for comparison. Their outer diameter and internal diameter were 38 mm and 46 mm, respectively, as shown in Figure 3b. Their thickness was 10 mm. They were polished before the test, resulting in a surface roughness of 200 ± 40 nm.

### 2.2. Experimental Apparatus and Wear Tests

The wear tests between the modified TPU ring-plates and ZCuSn_10_Zn_2_ ring-plates were carried out on a sophisticated plate-on-plate friction test device (CBZ-1 test machine, Haima Ltd., Wuhan, China) under seawater, as shown in Figure 3a. The seawater was made using commercial sea salt and pure water, with a mass content of sea salt of 3.5%. During the tests, the lower TPU polymer ring-plate remained stationary, while the upper ZCuSn_10_Zn_2_ ring-plate slid on the TPU surface in a rotational motion. A low speed (0.11 m/s) and high nominal pressure (1 MPa, normal force was 326 N) were set to investigate the effects of PTFE particles on the wear behaviors of TPU. A pure PTFE plate was slid against a ZCuSn_10_Zn_2_ ring-plate under the same test conditions as a reference. The collecting frequency of COFs and frictional forces was 1 Hz. A new set of modified TPU and ZCuSn_10_Zn_2_ ring-plates were set up to collect the real-time frictional vibration and noise behaviors using a B&K PULES Measurement System (4535-B-001, Brüel & Kjær, Denmark), as shown in Figure 3c. The test time was 2 h, and each experiment was repeated three times under the same test conditions.

### 2.3. Measurement Techniques and Procedures

The wear mass losses of modified TPU plates were obtained by checking their weights before and after testing using a high-accuracy analytical balance (BSA124S-CW, Shanghai Shuangxu Electronics Co., Ltd., Shanghai, China). Each specimen was measured three times under the same test conditions. Before weighing, the modified TPU plates were cleaned in pure water and dried for 24 h in an oven at 40 °C. The thermal gravity curve of composite materials was recorded in the temperature range from 20 °C to 1000 °C using a simultaneous thermal analyzer (STA449F3, Netzsch, Bavaria, German) at a heating rate of 5 °C/min. The surface topographies and energy-dispersive spectra (EDS) were recorded using a scanning electron microscope (SEM) (JSM-6701F, TESCAN, Brno, Czech). Images of the wear surface morphologies of modified TPU ring-plates and ZCuSn_10_Zn_2_ plates were recorded using an optical microscope (OLYMPUS BX51, Tokyo, Japan). The surface roughness of the testing specimens was measured using a CLSM (VX-X1000, KEYENCE, Ōsaka, Japan). The real-time frictional vibration and noise behaviors were collected using a B&K PULES Measurement System (4535-B-001, Brüel & Kjær, Denmark).

## 3. Results

### 3.1. Analysis of Coefficients of Friction

Figure 4 shows the COF behaviors of the pure PTFE, modified TPU ring-plates, and ZCuSn_10_Zn_2_ ring-plates under seawater (1 MPa, 0.11 m/s). The pure PTFE presented an excellent friction reduction, and its COF stabilized at 0.05 as shown in Figure 4a, in line with the literature [32,33,34]. On the other hand, the TPU with no PTFE particles showed a severe wear process; its COF fluctuated remarkably, decreasing in the initial stage, but increasing sharply throughout the wear process, eventually reaching a high level of 0.21, which was four times that of pure PTFE. It is reasonable to conclude that the vibration behaviors occurred due to the severe friction process. When TPU was modified with 3% PTFE particles, the COF was obviously reduced and stabilized at around 0.17, albeit with a high fluctuation amplitude. This phenomenon demonstrates that PTFE particles could improve the self-lubricating properties of TPU. Although the TPU modified with 6% PTFE had a large COF in the initial stage, it showed an obviously decreasing trend in the first 30 min, stabilizing at about 0.125, thereby presenting an evident reduction in friction. When the mass content of PTFE particles was increased to 9%, the COF decreased significantly and stabilized at 0.1, thereby reducing the COF by more than 50% compared to the TPU with no PTFE, indicating its good self-lubrication performance. Moreover, the COF fluctuated slightly, revealing its stability. Upon further increasing the content of PTFE particles to 12%, the COF presented a trend which was initially similar to the TPU modified with9% PTFE. However, the COF and its fluctuation amplitude gradually increased thereafter, reaching 0.18 at the end of the test. This indicates that the wear process between the friction pairs was gradually aggravated. Figure 4b showed the average COFs of the pure PTFE and TPU modified with different mass contents of PTFE. The average COFs decreased with the increase in PTFE content, reaching the minimum value at 9% PTFE. However, a further increase in PTFE led to a sharp increase in the average COF. In summary, because of its excellent self-lubricating properties, a suitable mass content of PTFE could significantly enhance the self-lubricating properties of TPU. However, excessive PTFE was not conducive to a further reduction in the COF.

### 3.2. Analysis of Wear Mass Loss

Figure 5 shows the wear mass losses of the TPU materials modified with PTFE particles at 1 MPa and 0.11 m/s under seawater. The wear mass loss of TPU with no PTFE was substantial, reaching 0.0078 g. However, the addition of PTFE reduced the wear mass loss of the modified TPU due to the reduction in COF shown in Figure 4. The wear mass loss of TPU modified with 9% PTFE was only 0.0045 g, exhibiting the best wear reduction (more than 40%). When the mass content of PTFE was further increased to 12%, the wear mass loss sharply increased, exceeding the wear mass loss of TPU modified with 6% PTFE. This indicates that the TPU suffered a severe wear process, in line with the increasing COF shown in Figure 4. In summary, PTFE could improve the wear resistance of TPU composite materials by enhancing their self-lubricating properties.

### 3.3. Analysis of Frictional Vibration and Noise Behaviors

The severe friction processes of polymer materials often induce vibration and noise, which severely affect the reliability, stability, and noise level of the friction system. Figure 4 shows the unstable wear processes of the modified TPU; to ensure fairness, the vibration signals for 10 s from 7000 s to 7010 s were chosen to investigate the effects of PTFE on the vibration behaviors of TPU, and the results are presented in Figure 6. Generally, the severe friction process of TPU (no PTFE) induced strong vibration behaviors, as shown in Figure 6(a1), and it presented the largest fluctuation amplitude from −1.9 m/s^2^ to 1.5 m/s^2^, which reduced the tribological system’s stability. However, the addition of PTFE could reduce the COF and weaken the vibration behaviors of TPU. Specifically, modification with 3% and 6% PTFE obviously reduced the vibration amplitudes of the TPU during the wear processes, as shown in Figure 6(b1,c1). Moreover, modification with 9% PTFE endowed the TPU with the smallest vibration amplitudes from −0.5 m/s^2^ to 0.5 m/s^2^, as shown in Figure 6(d1), presenting the best vibration reduction ability. These phenomena confirm that modification with PTFE was conducive to enhancing the vibration damping performance of TPU. However, a high PTFE mass content went against the trend of deceasing vibration amplitudes of the TPU due to a worsening wear process (see Figure 4 and Figure 5). 

The frequency-domain signals of vibrations could disclose the distribution characteristic of the main vibrational frequencies, further revealing how PTFE affected the vibration behaviors of TPU. Figure 6 shows the frequency-domain signals according to the vibration time-domain signals of TPU modified with different mass contents of PTFE. Strong vibration behaviors of TPU with no PTFE were coupled by waveforms at frequencies of 183 Hz, 369 Hz, 578 Hz, 1183 Hz, 1757 Hz, and 1939 Hz, which were all higher than those observed for TPU materials modified with PTFE, as shown Figure 6(a2). The introduction of PTFE changed the distribution characteristics of the main vibrational frequencies. Specifically, 3% and 6% PTFE weakened the amplitudes of the main vibration frequencies, especially at higher frequencies (bigger than 1000 Hz), while some of the main vibrational frequencies faded away. When the TPU was modified with 9% PTFE, only low-frequency main vibrations remained, which were also decreased, as shown in Figure 6(d2). Thus, the high-frequency vibrations disappeared, indicating that they were not induced by the introduction of PTFE. 

Frictional vibration is a key factor for inducing frictional noise. Frictional noise behaviors can be used to characterize the wear state of the rubbing pairs. This study collected frictional noises between the modified TPU composites and ZCuSn_10_Zn_2_ ring-plates. To ensure the reliability of the friction noise data, we collected the average noises of different modified TPU composites at three friction stages, i.e., during the first 10 min (range from 0 s to 600 s), during the middle 10 min (range from 3300 s to 3900 s), and during the last 10 min (range from 6600 s to 7200 s), in an effort to characterize the effects of PTFE particles on the frictional noises of modified TPU, and the results are shown in Figure 7. During the first 10 min, the average values of frictional noise of the TPU (no PTFE) and TPU composites modified with PTFE were close to each other, distributed within a range from 70 dB to 72 dB. The friction pairs may have still been in the running period, with relatively close wear states, as proven by the similar trends of COFs shown in Figure 4a. However, as the friction processes continued, there were obvious differences in the frictional noise values during the middle 10 min. The friction noise value of the TPU with no PTFE increased sharply, which was directly related to the severe wear process (see Figure 4). The friction noise value of the TPU modified with 3% PTFE also had an increasing trend, which conformed to the high COF during this period (see Figure 4a). When the TPU was modified with 6%, 9%, and 12% PTFE particles, the frictional noise values decreased slightly, because the wear conditions were significantly improved during the middle 10 min (obvious reduction in COFs). During the last 10 min, the frictional noise value of the TPU with no PTFE continued to increase, which was consistent with the rising trend of its COF. The friction noise value of the TPU modified with 3% PTFE particles still had an increasing trend, whereas the friction noise value of the TPU modified with 6% PTFE particles retained a similar value. The friction noise value of the TPU modified with 9% PTFE particles still exhibited a decreasing trend due to the decreasing trend of the COF shown in Figure 4a. When TPU was modified with 12% PTFE particles, the friction noise value increased sharply, indicating that a severe wear state occurred, which was related to the sudden increase in COF (see Figure 4). 

These results indicated a direct relationship between frictional vibration and noise and the wear states of the friction pairs. Large COFs and fluctuation amplitudes could induce significant frictional vibration and noise. PTFE particles improved the self-lubricating properties of the TPU and reduced the COFs, ultimately reducing the friction-induced vibration and noise. The introduction of 9% PTFE particles led to the best friction reduction and, thus, the best frictional vibration and noise reduction. The introduction of 12% PTFE particles was not conducive to long-term frictional vibration and noise reduction.

### 3.4. Analysis of Wear Surfaces

Figure 8 displays the optical photographs (500× magnification) of the surface topographies of the TPU modified with PTFE particles. Obviously, the tin bronze metal layers were transferred and adhered to the wear surface of the TPU during the wear process, indicating severe friction, as shown in Figure 8a. However, the TPU composite with 3%PTFE featured a smoother wear surface despite heavy deformations occurring in the area without PTFE distribution (see Figure 8b). Moreover, PTFE particles appeared on the wear surface and weakened the friction behaviors, thus leading to stable and smooth worn surfaces, especially with 6% and 9% content, as shown in Figure 8c,d. In addition, many PTFE particles were present on the TPU surface with 12% modification, but there were also ZCuSn_10_Zn_2_ particles, transferred from ZCuSn_10_Zn_2_ plates, distributed on the wear surface, as shown in Figure 8e. Therefore, excess PTFE did not contribute to a further reduction in wear. In general, the TPU composites modified with 6% and 9% PTFE fillers showed slight wear, which was consistent with the wear mass losses and COFs.

Figure 9 exhibits the wear surfaces of ZCuSn_10_Zn_2_ plates sliding against TPU composites. The ZCuSn_10_Zn_2_ plate showed noteworthy furrow scratches on the wear surface when it was worked as a frictional pair with the TPU with 0% PTFE (Figure 9a). Furthermore, it should be noted that the wear surface gained an obvious reddish appearance, which might have been caused by oxidation during the heavy wear process. However, there were no obvious wear scratches on the ZCuSn_10_Zn_2_ plate surface outside of slight cracks when in contact with the TPU with 9% PTFE, while it generally maintained its original color, suggesting less oxidation and a slight wear process, as shown in Figure 9b. Moreover, severe and significant wear scratches appeared on the ZCuSn_10_Zn_2_ plate surface in contact with the TPU modified with 12% PTFE, while a local reddish appearance also emerged (see Figure 9c). In addition, oxygen mass contents on the ZCuSn_10_Zn_2_ surfaces were detected, in order to verify the friction and wear processes, as shown in Figure 9d–f. The SEM image in Figure 9d demonstrates obvious cracks on the wear surface in contact with the TPU modified with 0% PTFE, while the EDS result in Figure 9e illustrates that the mass content of oxygen rose to 4.8% in the detected area, revealing significant oxidation due to the severe wear process on the ZCuSn_10_Zn_2_ plate. Figure 9f shows that, with the increase in PTFE content, the oxygen mass content on the ZCuSn_10_Zn_2_ plate exhibited a downward trend despite heavy oxidation taking place on the ZCuSn_10_Zn_2_ plate worn with 12% PTFE due to the severe wear and friction process.

## 4. Discussion

PTFE has a typical symmetrical linear molecule structure with no branches, and it possesses a smooth profile of molecule chains. Blanchet found that the smooth profile enabled the PTFE molecular chain to be easily rearranged in parallel to the friction direction, which reduced the frictional force between the PTFE molecular chains, thus greatly reducing the coefficient of friction [35]. Meanwhile, Tabor observed that the slippage phenomenon between PTFE crystalline flakes under yield stress could result in a low shear stress between the PTFE molecular chains, eventually leading to low friction [36]. Thus, when PTFE material slides against a solid object, the COF between the contact surfaces is usually at a low level, sometimes even less than 0.01 [37,38]. Therefore, PTFE particles with this low friction performance could endow TPU with good self-lubrication properties. When ZCuSn_10_Zn_2_ ring-plates slid against the TPU composites filled with PTFE, as shown in Figure 10a, the smooth molecular profile and slippage phenomenon of PTFE allowed the ring-plates to easily slide over the modified TPU, facilitating the good lubrication shown in Figure 10b. On the other hand, some PTFE particles were peeled off from the surface of the modified TPU material, moving to the contact zone between the ring-plate and modified TPU wear surfaces, which also improved the friction state. These phenomena greatly reduced the COFs and their fluctuation amplitudes (see Figure 4a), as well as weakened the frictional forces and actions between the friction pairs (see Figure 8 and Figure 9), which further enhanced the wear resistance of TPU (see Figure 5). Furthermore, low frictional forces (according to COFs) and small fluctuation amplitudes were beneficial to reducing the wear surfaces of the rubbing pairs, which decreased the frictional vibration and noise, as shown in Figure 10c. Accordingly, the vibration waveforms at high frequencies were not induced due to the significant reduction in friction by the introduction of PTFE, which led to a decreasing trend of frictional vibration amplitudes and frictional noise levels, as shown in Figure 6 and Figure 7. 

## 5. Conclusions

This study developed a novel polymer material whereby TPU was modified with PTFE particles. Wear tests of modified TPU specimens with ZCuSn_10_Zn_2_ ring-plates were carried out under seawater to investigate the effects of PTFE particles on the wear and vibration behaviors of TPU. This study found that the PTFE particles had a great influence on the frictional vibration and noise properties of TPU, thereby satisfying the demand for high reliability and safety of the ship shaft support system. The following conclusions could be drawn:(a)PTFE particles reduced the COFs and wear mass losses of the TPU composites, with 9% PTFE reducing the COF and wear mass losses by more than 50% and 40%, respectively.(b)A suitable mass content of PTFE obviously weakened the amplitudes of the main vibration frequencies of the TPU materials, as well as eliminated vibration waveforms at higher frequencies.(c)The lower COFs of TPU modified with PTFE resulted in a small fluctuation amplitude, which improved the stability of the tribological system, resulting in an ideal reduction in frictional vibration and noise.

## Figures and Tables

**Figure 1 materials-15-01365-f001:**
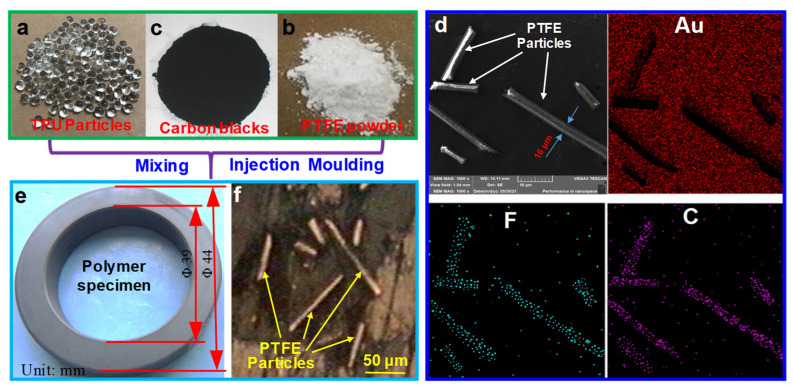
Preparation of test materials: (**a**) TPU particles; (**b**) PTFE powder; (**c**) carbon black powder; (**d**) micro-morphology of the PTFE particles and their distribution maps of fluorine and carbon; (**e**) modified TPU specimen; (**f**) microscopic optical photograph of modified TPU surface with 6% PTFE particles after polishing.

**Figure 2 materials-15-01365-f002:**
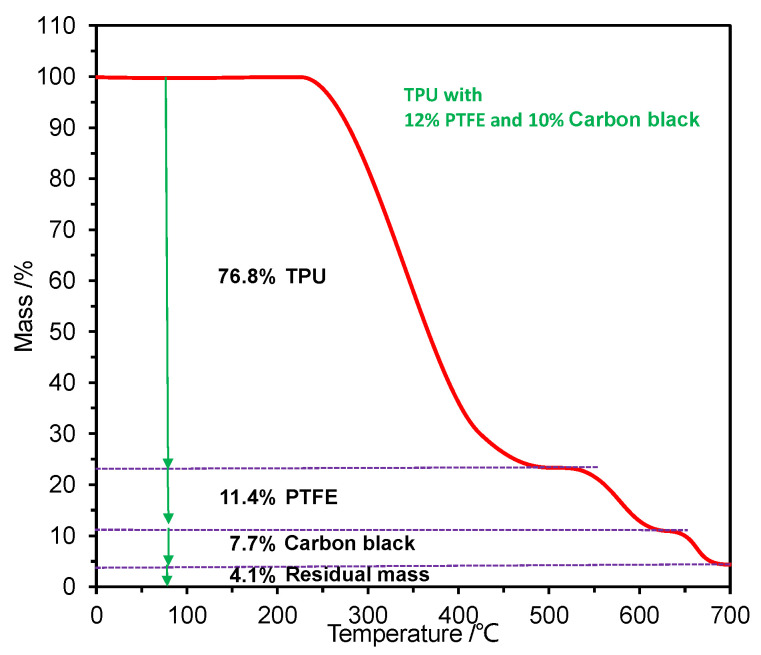
Thermal gravity (TG) analysis of modified TPU with 12% PTFE and 10% carbon black.

**Figure 3 materials-15-01365-f003:**
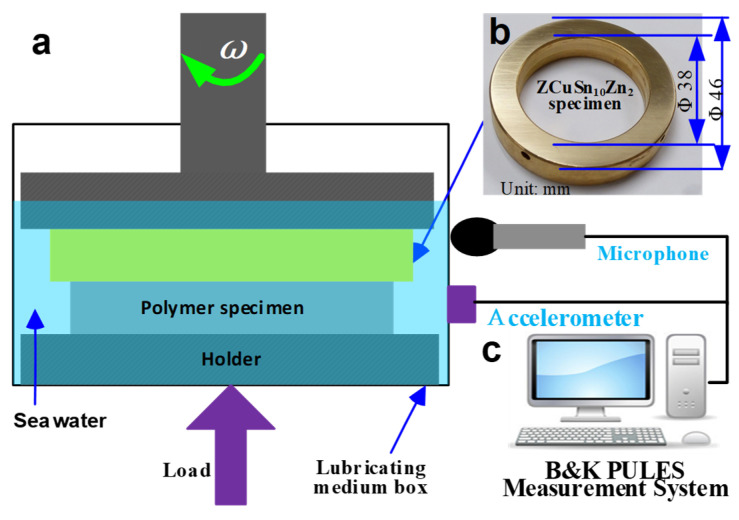
Test apparatus used in this study: (**a**) schematic diagram of experimental process; (**b**) ZCuSn_10_Zn_2_ ring-plate; (**c**) frictional vibration and noise collection system.

**Figure 4 materials-15-01365-f004:**
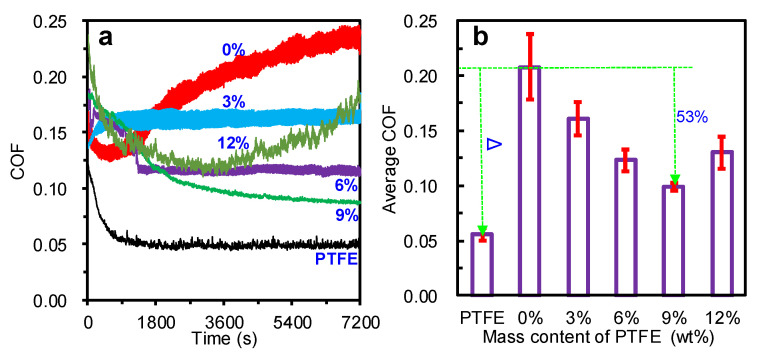
COF behaviors of pure PTFE, modified TPU composites, and ZCuSn_10_Zn_2_ ring-plates at 1 MPa and 0.11 m/s under seawater: (**a**) COF behaviors of pure PTFE and TPU composites modified with different mass contents of PTFE particles; (**b**) average COFs of the pure PTFE and modified TPU composites.

**Figure 5 materials-15-01365-f005:**
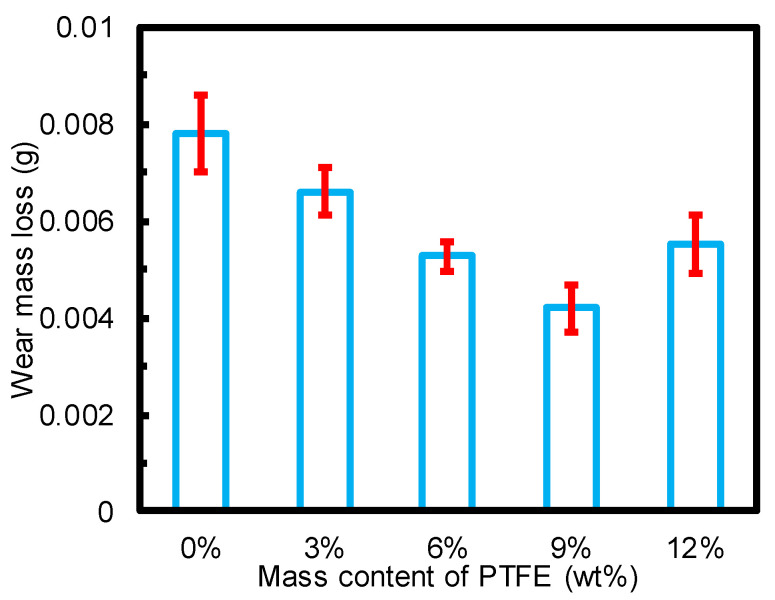
Wear mass losses of the TPU modified with PTFE particles at 1 MPa and 0.11 m/s under seawater.

**Figure 6 materials-15-01365-f006:**
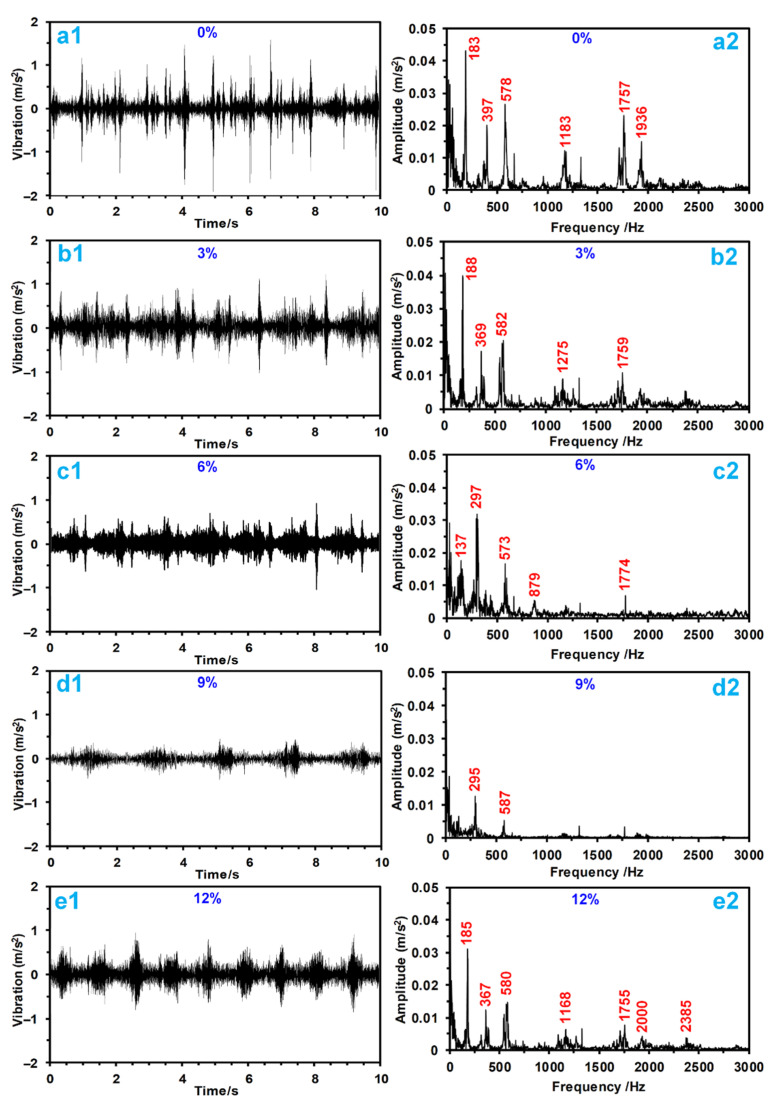
Vibration signals between modified TPU composites and ZCuSn_10_Zn_2_ ring-plates at 1 MPa and 0.11 m/s under seawater over 10 s from 7000 s to 7010 s. Vibration time-domain signals of TPU modified with (**a1**) 0%, (**b1**) 3%, (**c1**) 6%, (**d1**) 9%, and (**e1**) 12% PTFE; vibration frequency-domain signals of TPU modified with (**a2**) 0%, (**b2**) 3%, (**c2**) 6%, (**d2**) 9%, and (**e2**) 12% PTFE.

**Figure 7 materials-15-01365-f007:**
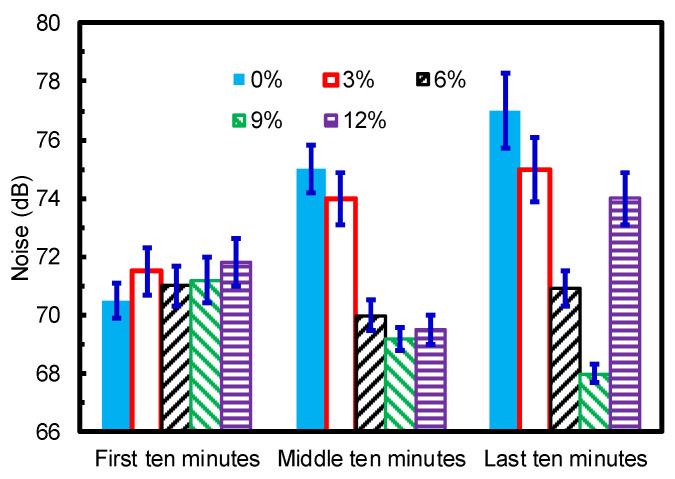
Frictional noise between the modified TPU composites and ZCuSn_10_Zn_2_ ring-plates at 1 MPa and 0.11 m/s under seawater.

**Figure 8 materials-15-01365-f008:**
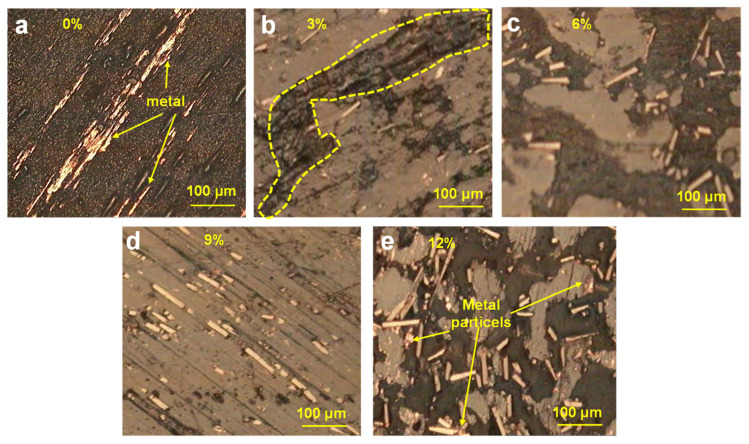
Wear surface morphologies of the TPU composites modified with (**a**) 0%, (**b**) 3%, (**c**) 6%, (**d**) 9%, and (**e**) 12% PTFE particles at 1 MPa and 0.11 m/s under seawater.

**Figure 9 materials-15-01365-f009:**
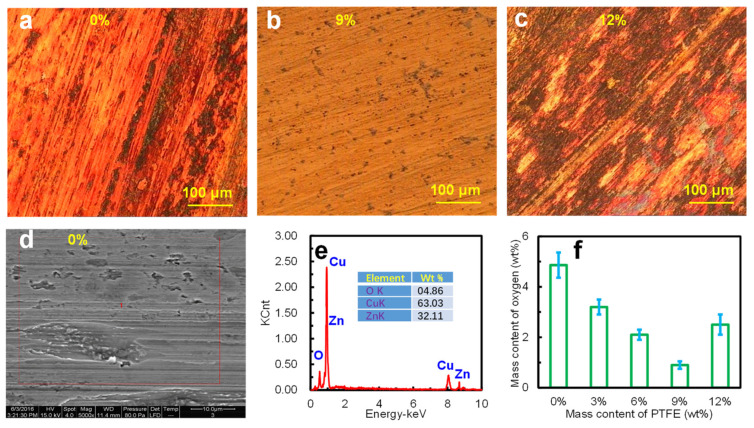
Wear surface morphologies of the ZCuSn_10_Zn_2_ plates sliding against TPU composites modified with (**a**) 0%, (**b**) 9%, and (**c**) 12% PTFE particles. (**d**) SEM image and (**e**) EDS of sliding surface of the ZCuSn_10_Zn_2_ plate worn against TPU (no PTFE). (**f**) Mass content of oxygen on the sliding surfaces of the ZCuSn_10_Zn_2_ plates worn against TPU composites with different mass contents of PTFE particles.

**Figure 10 materials-15-01365-f010:**
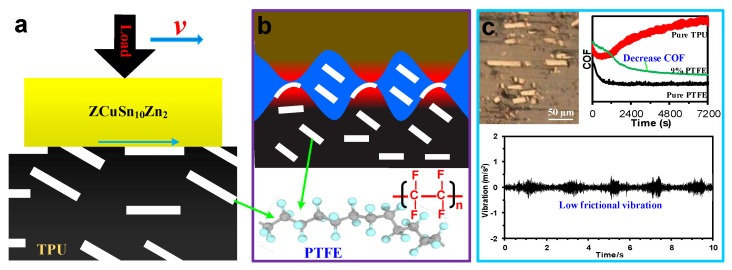
Frictional behaviors of TPU modified with PTFE: (**a**) contact diagram between the modified TPU and the ZCuSn_10_Zn_2_ ring-plate; (**b**) lubrication model of modified TPU sliding against ZCuSn_10_Zn_2_; (**c**) good lubrication behavior and excellent frictional vibration reduction performance.

**Table 1 materials-15-01365-t001:** Physical properties of TPU and PTFE.

	Shore Hardness (A)	Tensile Strength (MPa)	Elastic Model (MPa)	Elongation at Break (%)	Melting Point (°C)	Density (kg/m^3^)	Maximum Operating Temperature without Load (°C)
PTFE	65	27.6	280	238	327	2.19×103	260
TPU	85	35	400	350	190	1.18×103	140

**Table 2 materials-15-01365-t002:** Formulas of the modified TPU composite materials.

Modified TPU Composites	TPU with No PTFE	TPU with 3% PTFE	TPU with 6% PTFE	TPU with 9% PTFE	TPU with 12% PTFE
TPU	180 g	174 g	168 g	162 g	146 g
PTFE particle	0 g	6 g	12 g	18 g	24 g
Carbon black (N772)	20 g	20 g	20 g	20 g	20 g

## Data Availability

The data presented in this study are available on request from the corresponding author.

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
