# Peer review of "Reinforcement of Frictional Vibration Noise Reduction Properties of a Polymer Material by PTFE Particles"

_materials, 2022, doi:10.3390/ma15041365_

Round 1

Reviewer 1 Report

The article titled “Reinforcement of frictional vibration noise reduction properties of a polymer material by PTFE particles” appears very interesting by the highly important applied subject.

In general, the discussion and the conclusion section are well elaborated. However, the realm of the article that is the material fabrication is poorly described as to be replicated by the readers.

Additionally, the authors fail to describe properly some of the characterization sections, and some mandatory checking of the material is missing. Note that the related to the friction properties may be OK, but the absence of any control in many key aspects makes these to be not robust enough.

Below these lines some queries are listed:

  1. In the introduction section the sentence “A novel polymer material that the thermoplastic polyurethane (TPU) was modified by polytetrafluoroethylene (PTFE) particles was developed” is confusing. Please, rewrite it for readable reasons.
  2. Some additional properties of the polymers used, rather than the merely physical ones, must be provided. For instance, the molecular weight.
  3. Equally, the trademark, grade, and/or supplier of the starting materials must be provided.
  4. In the same way, the aspect ratio of the PTFE rods must be included in order to evaluate the reinforcement effect under robust arguments. Please, include the following including the dispersion measurements.
  5. In the same sense, please include the carbon black particle size and particle size distribution.
  6. The authors provide the formulas for the materials (with the nominal amounts of PTFE and CB) but the real content of both is not checked. Note that when using extrusion It very often occurs that the nominal dose of the fillers and the real one is very different. Please check it. A TGA may be enough.
  7. The processing parameters for the extrusion process must be provided. For instance: T profile from hopper to die, L/D ratio, gear rate, residence time, et cetera. In absence of these, the process cannot be replicated.
  8. The authors mention the following “PTFE particles could keep the original form”, but fail in providing pieces of evidence of that. This reviewer wonders if the authors have controlled this key aspect. The fact is that the images in figure 1 suggest that the figures have not the same dimension, neither width nor length. In other words, they exhibit different aspect ratios. Please, the authors must check the latter.
  9. Provide the conditions for the SEM microscopy (Voltage, tilt angle, sample preparation et cetera). Equally, provide the observation mode for the optical microscopy.
  10. ¿SEM over Sputtered samples or nor?
  11. Provide better images for F and C images in figure 1. They are not clear at all (at least in the manuscript I managed).

In the light of the above-mentioned concerns, the recommendation must be to perform a MAJOR revision since the material obtain and most of the basic characterization procedures are missing or poorly described to assure traceability (the realm of scientific communication).

Reviewer 2 Report

This is a nice contribution related to improving frictional vibration noise reduction properties of a polymer material by PTFE particles. I will recommend it for publication after several minor points are addressed as below.

  1. Line 59-61, 'Many scholars have used kinds of nanoparticles to enhance the self-lubricating property of polymers [18-20].' Several relevant studies (10.1126/science.aay8276; doi.org/10.1021/acsami.1c12631) should be included to support such claim.
  2. Is that possible to perform static analysis for the data presented in figure 3 and 4?
  3. It will be better if the authors could explain why ZCuSn10Zn2 ring-plate was chosen for the frictional measurements.

Round 2

Reviewer 1 Report

The authors have performed a good revision by considering the concerns of the previous revision draft. The modifications are enough convincing to recommend the publication of the article in its actual form.

Author Response

Comments of Reviewer

The authors have performed a good revision by considering the concerns of the previous revision draft. The modifications are enough convincing to recommend the publication of the article in its actual form.

Response: Thanks very much for your affirmation and encouragement. Your valuable comments and suggestions on my submitted paper are helpful to improve the quality of my paper. I have spent much time modifying the original manuscript again, including the terminology, mistakes, English writing, and grammar. Moreover, I have enriched the description of the experimental data of the article to make it substantial, and the overall structure of the article has became more complete. Thank you very much again.